# EvoDesigner: Evolving Poster Layouts

**DOI:** 10.3390/e24121751

**Published:** 2022-11-30

**Authors:** Daniel Lopes, João Correia, Penousal Machado

**Affiliations:** 1Department of Informatics Engineering, Centre for Informatics and Systems of the University of Coimbra, University of Coimbra, 3004-531 Coimbra, Portugal; 2Intelligent Systems Associate LAboratory (LASI), University of Minho, 4800-058 Guimarães, Portugal

**Keywords:** automatic, evolutionary, graphic design, layout, grids, posters

## Abstract

Frequently, one of the goals of Graphic Design (gd) is discovering disruptive visual solutions that stand out and attract people’s attention. However, due to the increasing democratisation of gd, graphic designers tend to adopt design trends, leading to designs that many times lack innovative and catchy features. *EvoDesigner* is an evolutionary extension for *Adobe InDesign* that aims to aid gd processes by automatically evolving layout and style variations of given *InDesign* pages. The generated pages might be previously created and post-edited by designers, promoting co-creation. As an extension of the study *EvoDesigner: Towards Aiding Creativity in Graphic Design*, this article begins with a general introduction of *EvoDesigner*. Then, we review previous experiments on evolving pages towards the page balance of existing target posters. Furthermore, we present new experiments exploring the benefits of using grid systems to position and scale page items along with a user survey made to gather feedback about the impact of grid systems in the generated pages and showcase examples of artefacts created from the collaboration between designers and the system. The findings indicate that the presented techniques can be used to interpret current layouts in different manners, and suggest that grid systems may be a useful tool for promoting the automatic production of layouts with better organisation when compared to applying no organisational constraints. However, a conducted user survey indicates that, depending on the goals of the designers, more organised layouts might not always be synonymous with better results.

## 1. Introduction

Graphic Design (gd) artefacts may have different goals according to the context of their applications. For example, these might be (i) communication artefacts, to pass information objectively to a given public or (ii) artistic artefacts, to pass information in a non-objective way, i.e., that is susceptible to personal interpretation or simply to be aesthetic. Either way, gd artefacts often must manage to attract the attention of the respective target public, and only after that may the public enjoy their aesthetics or read their information.

Enhancing aesthetics is an approach commonly used by designers for making designs stand out over competing ones (e.g., from other posters nearby or other book covers on the shelves) and therefore attract people’s attention. Nonetheless, as gd becomes more and more democratised, broadly produced, and shared (e.g., through social media), many designers tend to adopt trendy visual solutions, frequently leading to common artefacts that might not attract attention as effectively.

Seeking innovative solutions, graphic designers have always explored new technologies. Nowadays, we can observe the increasing exploration of digital techniques, such as digital animation and coding, to create moving and interactive designs able to stand out over static ones [1]. Nonetheless, such approaches might not be possible in all contexts due to technical issues and because developing such digital artefacts can be time-consuming and expensive. Therefore, creating disruptive aesthetics is often be a fundamental gd task when designing static, dynamic, or interactive artefacts. We believe that computational tools for aiding the exploration of unexpected gd ideas can help designers to achieve more distinctive aesthetics and free them up to explore and innovate in other matters, for instance, by creating dynamic or interactive designs or even creating brand-new features that are unimaginable at the moment.

To test the aforementioned belief, we present *EvoDesigner*, a system to automatically evolve an undetermined number of text boxes, shapes, and images into two-dimensional pages (canvases). The system was developed as an installable extension for *Adobe InDesign*, a broadly used gd software for desktop publishing, to make it easier for users to apply the generated ideas in professional environments. Moreover, in this way, human designers and *EvoDesigner* can alternately collaborate on the same designs by editing and evolving them using the same software.

In further versions of *EvoDesigner*, the system must be able to evaluate pages regarding their balance, legibility, and innovation degree. However, this preliminary iteration aims only to validate the functioning of the evolutionary engine and the inherent crossover and mutation methods. Thus, fitness is assessed through an existing and well-stated metric. More specifically, to test the system, we try to approximate the layouts of given target images by calculating the Mean Squared Error (mse) between the generated individuals (pages) and either sketched or camera-ready posters, hopefully achieving layouts that are similar to the targets’ as well as relatively unexpected.

In previous experiments [2], posters were generated with no organisation constraints, making retrieving layouts composed of aligned page items or other types of visual organisation almost impossible. In the present experiments, inspired by well-stated human design processes [3], we test the impact of grid systems in promoting more organised layouts, then compare the old and new results regarding fitness and phenotypes.

Primarily, in this extension of our previous paper [2], an overview of *EvoDesigner* is first provided. Then, we describe the evolutionary engine mentioned above. Lastly, we present both previous and current experiments for technical validation of the system and draw conclusions from a comparison of the two. The results suggest that the system can be viable for evolving gd artefacts within *InDesign* and demonstrate the feasibility of manually contributing to the designs. Furthermore, mse is revealed to be satisfactorily effective for chasing layouts that resemble given target posters, e.g., for evolving *InDesign* pages towards drafts of poster layouts. Lastly, feedback from designers gathered via an online user survey indicates that grid systems might aid the promotion of more organised layouts. Thus, these might be worth using during the evolution of gd posters whenever designers seek better structure rather than more expressiveness.

## 2. Related Work

This work intends to contribute by proposing (i) an evolutionary tool for assisting the development of two-dimensional gd artefacts such as posters or book covers, (ii) a tool that can be rapidly integrated into the workflow of graphic designers, and (iii) a tool that makes use of the editing capabilities of existing desktop publishing software. Thus, *EvoDesigner* refers mostly to computational systems that produce page layouts, including the design and geometry of the page items. This section highlights characteristics that served as inspiration for our system and lists the advantages and disadvantages of comparable works.

Generative approaches have become increasingly popular for developing gd artefacts. Such systems often use sets of predefined constraints to semi-randomly define visual features such as items’ colour, size, or position [4]. Nonetheless, generative designs can often be relatively predictable. Thus, these systems are more frequently used for generating design variations within predefined brand identities, rather than exploiting innovative ideas. For example, they can help create variations of book covers [5,6,7,8] or logos within the same brand identity [9,10].

Even so, we were able to pinpoint a few general-purpose generative projects; for example, for aiding the generation of typography [11,12,13]. More relevant to our project, we highlight generative projects for creating gd layouts. While several projects are more limited in terms of functionalities, e.g., by not allowing the setting of concept-wise parameters [14,15], others, such as the work of Ferreira et al. (2019) [16] or Cleveland (2010) [17], stand out by allowing the user to freeze certain intended parameters and allow the system to vary others. In this way, the user is able to preserve an intended style. Further *EvoDesigner* exchanges must use a similar strategy. Additionally, we highlight the work of Rebelo et al. (2020) [18] on creating web pages based on the semantic characteristics of their inner text content. In addition, it is necessary to implement methods for generating designs that can visually represent given semantic concepts. Lastly, we highlight the work of Feiner (1988) [14] and Cleveland (2010) [17], which make use of grid systems to generate layouts.

In addition to generative approaches, researchers have applied intelligent approaches [19] for generating visual artefacts, for example, using Machine Learning (ml) models to learn existing artefacts and then generating interpolations of these by exploring their latent space. In gd, among other uses, this approach has been used to generate logos [20] and typography [21,22,23,24] and applied to image editing [25,26]. More related to our project is the work of Zheng et al. (2019) [27] on creating content-aware layouts.

The shortcoming of ml approaches is that they often lead to imitation of existing styles [28]. Hence, while ml might be useful to interpolate existing artefacts, such techniques might not be the best for exploring more disruptive aesthetics. We argue that Evolutionary Computation (ec) can have greater potential in this regard, as such technique might often resemble human design workflows [29], i.e., either human designers or ec systems can explore the possibilities’ space towards a given conceptual target. This is frequently a relevant ability, as gd projects often have a briefing to respond to. In this sense, ec can be useful for complementing design processes by presenting designers with new ideas as well as by allowing experimentation with dozen or hundreds of designs in a relatively short period. Nevertheless, human designers are essential in the process of curating the generated results, fine-tuning them, and other complex tasks, such as extrapolating the visual ideas into different gd applications and adding dynamic and interactive features to the designs.

If not the majority, then at least a significant number of ec systems for gd applications use interactive methods which need the user to direct the evolution process. In this regard, work has been done on the creation of figures [30], icons [31,32,33], logos [34,35], typography [36,37,38,39], websites [40,41], and posters [42,43]. Among the projects we reviewed in this area, the work of Önduygu (2010) [44] might be the most complete, as the author evolves a considerable number of different features, specifically, typefaces, lines, shapes, colours, images, and visual filters. We intend to broaden this set of functionalities in *EvoDesigner* in order to pair the system as closely as possible with human designers. Ultimately, the system must be able to edit as many features as human designers can within the *Adobe InDesign* environment.

Less work appears to have been published concerning automatic ec creative systems. This may be due to the difficulties in creating evaluation metrics to objectify aesthetic qualities. Therefore, even though a few frameworks have been put out [45,46,47,48], none of them are able to resolve the aesthetics evaluation problem. However, there are automatic ec systems for gd applications that we may name, such as that of Rebelo et al. (2017) [49] for evolving moving posters based on the actions of bystanders.

Moreover, we refer to the work of Rebelo et al. (2018) [50], which mixes automatic and interactive evaluation metrics, allowing designers to collaborate with the systems during the evolutionary process.

Using ec systems to explore the space of possibilities and ml for determining fitness may be another useful strategy. There are numerous projects in the area of computational art [51,52,53]. However, there may be fewer references filed with respect to gd. Nonetheless, relevant works can be mentioned, such as that of Martins et al. (2016) [54] for evolving entire fonts out of provided modules. More specifically, the designer must provide the system with a number of modules (e.g., ones that are pertinent to a given project); the ml model then drives the evolutionary process towards a readable typeface.

Last but not least, we look at previous work that blends ec into pre-existing desktop publishing software. Based on our assessment, such integrations do not appear to happen very frequently. One of the few examples we could find is *Microsoft PowerPoint’s Design Ideas* [55]. The latter can be a good analogy for the workflow in *EvoDesigner*: (i) the system is integrated into a widely used software program; (ii) it takes advantage of the built-in functionalities; (iii) the users must start by inserting content, and the system then suggests possible layout and style solutions; (iv) the user and the system can both contribute to the final results; (v) the user can post-edit the results. Furthermore, we refer to *Evolving Layout* [43] due to its integration as an extension for *Adobe InDesign*, such as *EvoDesginer*. The shortcoming of *Evolving Layout* is that it only allows for interactive evolving of a reduced number of visual features, i.e., the position, scale, and rotation of the page items. As mentioned before, our system aims to evolve designs automatically as well as to edit a wide range of features on par with human designers.

## 3. Approach

We propose *EvoDesigner*, an automated evolutionary system for evolving pages, to help with the development of innovative gd solutions. The system must support graphic designers during the design experimentation phases to allow both human and machine participation in the process by editing pages (individuals). To do this, *EvoDesigner* is provided to users as an extension (plugin) for the popular desktop publishing software *Adobe InDesign*; html, css, *JavaScript*, and *ExtendScript* (*JavaScript* for *Adobe* software) were used to develop the built-in extension.

The main module of *EvoDesigner* uses an evolutionary engine based on a conventional Genetic Algorithm (ga) [56] with automated fitness assignment. Nevertheless, the system as a whole may be described as the combination of many modules: (i) the aforementioned evolutionary engine; (ii) several modules for visually evaluating images that may or may not be chosen by the user to assess fitness, such as modules for evaluating the degree of innovation, legibility, balance, relatedness to a given gd style, or even similarity to a given image; and (iii) a module that converts user-defined keywords into visual features (e.g., colours, geometric transformations, font weights, and others). The last module might be helpful for initially narrowing the search field in order to produce results that are more visually associated with the concept (keywords) of the various projects.

The present improvements involve putting the mentioned evolutionary engine to use and evaluating it while employing an image-similarity measure to determine fitness. Therefore, this paper does not include advances in the innovation, legibility, balance, or style evaluation modules. Nevertheless, Figure 1 presents a complete schematic depiction of the system. In addition to allowing the built evolutionary engine to be validated, using image-similarity measures for assigning fitness may be useful in real gd tasks, for example, for identifying surprising layouts that match supplied drafts which are used as target images in most of the experiments detailed further in this article.

However, merely placing objects around where they are supposed to be on a poster may not be sufficient to achieve aesthetic harmony. Because of this, designers frequently employ grid systems to position objects in more orderly ways, e.g., by aligning them [3]. As a result, expanding our previous experiments, a grid system was built to position and scale page objects accordingly [2].

To get started with *EvoDesigner*, as is customary when beginning a project in *InDesign*, the user must first create a blank document and insert the desired items (i.e., text boxes, images, or shapes) into the pages. The system variables can then be configured via a user interface (see Figure 2). The following variables are currently permitted: (i) the number of generations to be run, (iii) the population size, (iii) the number of pages to evolve, and (iv) the items that must always be present on any page. Additional functionalities will be made possible in future developments, including (i) inserting keywords, (ii) specifying desired or wanted visual features and edition tools (such as particular colours or typefaces), (iii) defining the visual hierarchy of the items, and (iv) selecting the appropriate fitness modules and their relative importance to the user. The user must click the "Generate" button to launch the system after the system options have been configured. When the evolution is finished, the user can edit the produced pages using *InDesign* or evolve the pages again.

### 3.1. Evolutionary Engine

Before starting the system, the user must specify which pages (individuals) should be evolved, from one to any number of pages. For instance, the user might only want to evolve three pages out of a ten-page document. The user must choose the target population size. If necessary, the population automatically increases to fit the number of specified pages. However, if there are fewer chosen pages than the specified population size, the system automatically produces the missing individuals by crossing over and mutating the chosen pages. Additionally, it confirms that each page contains all the mandatory items.

For the system to know which objects must be treated as equivalent, the user can label the items. For instance, even if two or more pages have items with the same name but distinct visual styles, the system treats them as equivalent. However, in the current version of the system, naming is only useful for mandatory items, as assigning any item the name of a mandatory item makes the other item mandatory as well. Therefore, the mandatory item criteria are met as long as one of these same-named items is present on the page.

Following initialization, the engine evaluates the individuals (pages). It then checks for termination criteria, which may include (i) finding an individual whose fitness equals or exceeds a given value (which is not taken into consideration at this point), (ii) determining whether the system ran a certain number of generations, or (iii) determining whether the user instructed the system to stop evolving by clicking the “Stop Generation” button.

If none of the termination requirements is met, selection is carried out using a tournament method with a size of two and an elite level of one individual. Finally, a new population is produced through crossover and mutation, the offspring are assessed, and the process is repeated.

#### 3.1.1. Representation

Phenotypes consist of the native rendering of *InDesign* pages, which can include a variety of items such as text boxes, shapes, or images. Items themselves are then described by a variety of positioning, shape, and style characteristics. *InDesign* automatically saves these properties in json format. Accordingly, genotypes in *EvoDesigner* are made up of json objects that have all the attributes of the corresponding pages as well as the properties of the items contained in them (see Figure 3 for a schematic example of the genotype). However, only the following item properties were taken into account in the current experiments: the surrounding box’s shape, size, position, order (z-position), flipping mode, blending mode, opacity, background colour or gradient, background tint, stroke colour or gradient, stroke weight, rotation, and shearing angle. For text boxes, extra features include text size, typeface, justification, vertical text alignment, letter spacing, and line height. Additional properties will be included in further developments. In addition, the parameters “name” and “label” are only used to monitor mandatory items and do not affect phenotypes.

#### 3.1.2. Variation

All individuals are subjected to the processes of crossover and mutation. In the current iteration, crossover only affects whole items; individual item characteristics are unaffected. The crossover procedure works as follows: we randomly iterate through each item in the first parent (p1) (i.e., items are not picked based on their order on the page). There is a 50% chance for each of these i1 to be passed on to the next generation with the same position, geometry, and style. If this is not the case, the system attempts to choose a random item i2 from parent 2 (p2) that has not yet been passed on to the offspring. If there is no such i2, i1 is passed. Otherwise, i2 is passed in its place. As a result, the minimum and maximum number of items in the offspring are respectively equal to the number of items in the smaller and larger individuals of the starting population (which, as mentioned before, might be automatically generated by the system).

Mandatory items can only be substituted with equivalent mandatory items, i.e., if i1 is mandatory, then i2 must have the same name as i1. Otherwise, i1 is passed. This constraint can be utilised to, for instance, restrict the shifting of text boxes containing titles (i.e., items named “title”) to other text boxes containing titles. Similarly to the project *Ȧdea* [57], we refer to such an approach as topological crossover when the shifts happen among similar structural components. For further clarification, if an item named “title” is set as mandatory, the offspring invariably inherits the name “title” from one of the parents. An example from nature is always acquiring important structural components, such as eyes, from either the father or the mother. This is important in our system, as it ensures that each poster contains at least the necessary (structural) components. It is more challenging to identify a natural analogue for optional items; however, for the purposes of example, one might consider it as inheriting (or not) a skin sign or a chronic disease.

Each mutation process may occur for each individual with a 1% chance of modifying one of the position, geometry, or style characteristics mentioned in Figure 3. Whenever mutated, each property receives a random value. These may be randomly chosen integers, floats, arrays of numbers, or constants from predefined lists. This applies to colours as well, which are selected from a predetermined set of colour values. Seven fixed colours are used in the current version: black, white, magenta, yellow, red, green, and cyan.

For experiments using grid systems, the items’ position and size are frequently constrained to a given grid. However, it is possible that this is not always the case, as mutation operations such as rotation and skewing can cause objects to be moved away from the grid guides. Page grids may be inherited from the parent pages, created manually by the user, or generated automatically using random parameters. These parameters include: (i) random top, bottom, left, and right margin sizes, which may be equal, different, or mixed sizes; (ii) a random number of columns and rows; and (iii) a randomly sized gutter between them. Examples of potential page grids are provided in Figure 4.

#### 3.1.3. Fitness Assignment

In this system iteration, fitness is determined by calculating how closely the created individuals resemble a particular target image. The pages are first exported in 72 dpi in the png format. These settings are shared by the target pictures. Then, using the mse, each page (in png) is compared with the provided target image, yielding a value *m* that represents the difference between the images. Therefore, the ultimate fitness value equals negative *m* for returning a similarity value (with higher values being better).

A well-established image similarity metric such as mse was selected to determine whether the created evolutionary engine could evolve properly. However, a metric of this kind might be useful for practical gd tasks as well. For instance, one might generate relatively unexpected layouts by approximating the page balance of provided images (either sketches or camera-ready images ) using given page items. Other fitness functions will be established in subsequent iterations, as previously noted. For instance, based on our experience with gd, we consider that evaluating innovation and balancing values may be crucial for defining disruptive and appealing gd artefacts. However, the latter (together with mse) might only be sufficient to produce more artistic artefacts, in which legibility might not be important. As a result, in order to generate communication design artefacts, a value for legibility must be retrieved and taken into account when assigning fitness. In addition, it may be useful to retrieve a second value that indicates whether an individual belongs to a specific gd style or aesthetic movement.

## 4. Experimental Setup and Results

The creation of gd posters is anticipated to be one of the main use cases for *EvoDesigner*. As a result, three hypothetical posters from Figure 5 (manually constructed from blank pages and only lightly stylised) were chosen to be evolved, initially with no organisation constraints and then utilising grid systems. A sample initial population of ten individuals (see Figure 6) was created from the same three pages of Figure 5 using crossover and mutation procedures.

The images in Figure 7 were used as targets. Figure 7b presents speculative posters that were manually created in *InDesign*. These were initially used to determine whether the system was evolving and up to what point. Figure 7a presents sketches representative of the corresponding posters in Figure 7b. These sketched posters were chosen as the targets for the major experiments, as they better demonstrate a use case for mse in fitness assignment. For example, an abstract layout might be sketched by a designer, and the system might then produce posters that resemble it using a specific set of page items. However, for various use situations, posters such as those in Figure 7b might be helpful as well. For instance, if a designer loves an existing poster, the system might assist in producing new ones with a similar page balance. Despite this, they must nevertheless differ sufficiently from the target ones after the specified page items have been altered in terms of position, geometry, and style.

The remaining settings were set as follows: (i) population size: 50; (ii) tournament size: 2; (iii) elite size: 1; (iv) probability of a page item crossing over: 50%; (v) probability of a mutation procedure to execute: 1%; (vi) mandatory items: all text boxes on the selected pages (the pages of Figure 5); (vii) fitness assignment: mse; (viii) termination criteria: achieving the defined number of maximum generations; (ix); grid system enabled: depending on the experiments.

The system was first run once without grid systems and with the parameter “maximum generations” set to 1000, in order to determine whether the fitness values were maximised and how many generations would be required before no significant advances were made. The poster in Figure 7b.1 was chosen as a target because it seemed to be more easily achievable among the presented posters, having visually heavier top-left and bottom-right corners (black items) and medium-weight items on the top and bottom sections (red stripes). This run was manually terminated at the 480th generation because there had not been any significant improvements for many generations; see Figure 8a, which shows the plotted fitness of the best individuals from each generation along with the average fitness of each population. As a result of this experiment, subsequent runs were set at 100 generations.

The following experiments were carried out without the use of grids by targeting each of the posters in Figure 7a while keeping the previously indicated settings. Four runs were produced for each poster. Figure 9 shows pages that the system produced for each of the target posters in Figure 7a. The average fitness of the best individuals from each generation for each target image is shown in Figure 10.

The resulting phenotypes indicate that the system was able to approximate the layout (balance) of the target images, as darker regions in the target images tend to result in more filled and darker areas in the created posters. Even the colour scheme, especially for darker colours, tended to be approximated.

As a result, it might be argued that the suggested method can already produce new posters utilising the page items that are provided by roughly replicating, while not exactly copying, the intended images’ layouts, i.e., by coming close to the target images without coming too close.

As previously indicated, it is visible from Figure 9 that the approach described above may not be able to build organised layouts such as those frequently created by human designers. In this regard, the subsequent experiments were carried out to determine whether grid systems could assist in the production of better-organised layouts, such as by frequently promoting the alignment of items with respect to others.

The same target page from Figure 9a was the target in the subsequent studies, except that the positioning and sizing of page elements were performed using grid systems. Using this configuration, 30 runs of 100 generations were produced. Figure 11 compares the average fitness of the best individuals using grids (from current experiments) and without grids (from earlier tests).

Although the introduction of grid systems may have negatively impacted fitness evolution, the fitness loss appears to not have been too severe, as the system was able to reasonably approximate the target layouts. Moreover, significant visual changes can be seen in phenotypes (see Figure 12), as page items are more frequently aligned to each other, promoting more organised layouts.

The inability to keep mandatory items visible is a foreseeable drawback of utilising mse for fitness assignments, regardless of the use of grid systems (ideally, mandatory items must be so whenever designers want them to be visible, rather than hiding behind other items, being off-page, or being too small to be seen). In this regard, incorporating the legibility assignment into the fitness calculation may be a viable strategy for enhancing outcomes in relation to this problem. However, designers might work around this problem by post-editing the outcomes to transform them from merely artistic into communication artefacts.

In this regard, Figure 13 shows posters generated before and after manual post-editing in order to demonstrate the potential of collaboration between human designers and the system.

## 5. User Survey

A user survey was conducted to test our assumption that the implementation of grid systems promoted more organised layouts (new experiments) compared to using no organisation constraints (previous experiments). As *EvoDesigner* is a tool developed for graphic designers, the survey was only conducted among design students, teachers, and professional designers. The answers were collected through an online questionnaire.

First, the users were asked for personal information in order for us to be able to detect possible biases or draw possible patterns among different respondent profiles. More specifically, users were asked about their age, occupation, level of design expertise and whether they worked or studied at the University of Coimbra. Thirty people responded to the questionnaire. The respondents’ ages ranged from 20 to 36 and they were 25 years old on average; 22 were students, of whom nine were in a bachelor’s degree program, seven a Master’s program, and six were PhD students. The remaining eight respondents were professional designers. Due to accessibility reasons, at the time of the survey most were or used to be somehow affiliated with the University of Coimbra in Portugal.

For the main questions, the respondents were presented with two sets of posters; set A showcased the generated posters from Figure 9 (i.e., excluding the target posters), while set B showcased the generated posters from Figure 12. Both sets of images were disposed of in a 6 by 2 matrix.

The first question aimed to understand which posters from set A or B seemed to be more organised. The following question, with an open answer, aimed to understand why. The next questions aimed to understand whether more organised layouts were helpful in achieving more finalised artefacts (which can be interpreted as more interesting results). Thus, the third question asked which set of posters, A or B, seemed to be in a more advanced stage of development. The fourth question again asked the respondents to explain the reason why.

Most respondents (28 out of 30) answered that set B was more organised than set A, which supports our assumption, i.e., that using grid systems to place and scale page items during the evolutionary process helps to promote more organised page layouts. The respondents in favour of set B argued that the posters were simpler, and had recognisable structure and recognisable alignment (to each other or to grid guides). In addition, a few respondents recognised a better pattern among posters, i.e., that the items were typically placed in the same zones among posters. However, this observation must be disregarded, as the posters from set A concerned three different target layouts. Furthermore, a few respondents argued that the posters from set B showed better legibility and hierarchy. Although no legibility and hierarchy metrics were utilised, we can infer that keeping text boxes in a grid helped to maintain the text on the page, making it more legible, unlike the posters in set A, in which the text boxes often left the pages regardless of the target layout.

Concerning the third question, opinions were split; however, they tended to be the other way around. Nineteen respondents thought that the posters from set A were in a more advanced stage of development, and eleven thought otherwise. Such results might indicate that achieving more organised layouts is not always of interest to graphic designers. In this sense, while it might be useful for grid systems to exist in creative systems for gd, grids must be left optional or be used only when the designer considers such organised aesthetics to be pertinent. When asked for the reasoning behind their answers, the respondents in favour of set B often argued these posters were more expressive or seemed to have more items. Taking the number of items on the page as an argument can indicate that these respondents (at least three) did not consider organisation as a key feature in answering the question. Nevertheless, arguing for expressiveness seems to be a valuable insight, i.e., not aligning items to a grid might help to achieve more expressive posters, while aligning them might help otherwise. Further supporting this assumption, two of the respondents indicated that this answer could depend on the context of the project. Both of these respondents selected set B in the third question. Other respondents answered that the posters in set A were more diverse. However, again, this argument must be disregarded, as the posters in set A concerned three different target layouts and the ones in set B did not. For this reason, similarity might not be a fair comparison argument. The respondents in favour of set B often argued that these posters seemed to be more thoughtful and organised. From the latter answers, it may be inferred that the respondents considered the task of organising and aligning items to be more laborious. Nevertheless, as already mentioned, placing items with no hard organisational constraints might help promote more expressive layouts.

## 6. Conclusions

Finding disruptive aesthetics in gd is typically of the utmost importance for attracting the target public’s attention. Designers frequently adhere to graphic trends, though this can produce work that may not be distinctive from competing gd artefacts.

*EvoDesigner* is a system that automatically evolves pages within the *Adobe InDesign* environment. Its goal is to assist in creative processes by alternately working with graphic designers to edit pages and page items, such as for making posters or book covers.

This article, an extension of our previous paper *EvoDesigner: Towards Aiding Creativity in Graphic Design* [1], presents a preliminary iteration of *EvoDesigner* consisting of the implementation and testing of an automatic evolutionary engine based on a conventional ga. Until now, experiments have been conducted utilising the mse as a fitness metric to evolve page layouts toward specific target images. To accomplish this, the created pages are exported from *InDesign* in the png format and compared to the provided target images, again in png. Additionally, we carried out comparisons between posters evolved with and without organisational constraints, more specifically, by either allowing items to be sized and positioned randomly on the pages or positioned and scaled in relation to page grids.

Target images for the current experiments included speculative posters of two types: (i) layouts that had been sketched up, and (ii) posters created using *Adobe InDesign*. Sketched targets may be useful, for instance, when a graphic designer wants to produce artefacts that approximate a specific colour scheme and page balance. However, using images of final gd artefacts can be useful as well, for instance, by simulating the targets’ page balance without producing outcomes that are overly identical to the originals.

The conducted experiments revealed that the proposed approach is workable in the evolution of gd artefacts that resemble the page balance of the target images and are unique enough to not be taken as replicas. We believe the proposed approach can be useful in the gd workflow for assisting in the creation of innovative gd solutions, with the system can accepting layouts and considering these in order to dispose and edit page items in relatively unexpected ways, at least in the creation of posters. However, further feedback from professional designers would be beneficial.

Furthermore, both the experimental results and the analysis of the feedback gathered from the conducted user survey suggest that positioning and scaling page items according to grid systems can aid in the generation of more organised layouts. Nevertheless, the user feedback suggested that generating more organised layouts might not always be desired, as the organisation may sometimes contradict expressiveness and different designers in different contexts might seek more of one or another. In this sense, grid systems might be worth implementing in co-creative computational systems; however, we suggest that these be made optional or used only when appropriate.

Our future work will involve the development of a number of different modules to increase the system’s robustness, such as (i) a module for converting keywords into visual properties or tools, for example, to narrow the search space to a specific creative concept, or (ii) fitness modules that can or cannot be used to perform innovation, legibility, and balance evaluations or determine how much an image may be in style with a specific gd aesthetic movement. Moreover, we must further test the system using larger populations and novelty search mechanisms to better promote diversity during evolution.

## Figures and Tables

**Figure 1 entropy-24-01751-f001:**
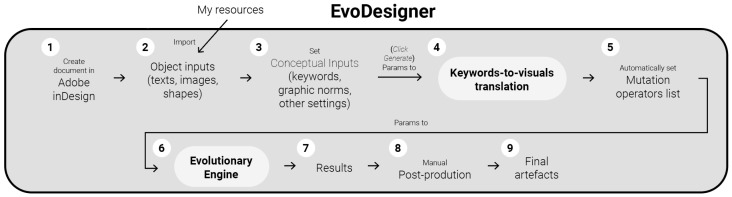
*EvoDesigner* depicted schematically. (1) The user must first create a blank document, then (2) insert elements into the pages, (3) set preferences (such as choosing which pages will be evolved and inserting keywords), and then click “Generate” to begin. (4) The system then attempts to find properties that match the inserted keywords, (5) each property is then assigned a probability of being used to mutate pages, and (6) the pages are finally evolved. (7) The output pages are made available as regular *InDesign* pages that (8) may be edited and (9) exported by human designers. The evolution can be restarted by altering the settings from any of the aforementioned stages.

**Figure 2 entropy-24-01751-f002:**
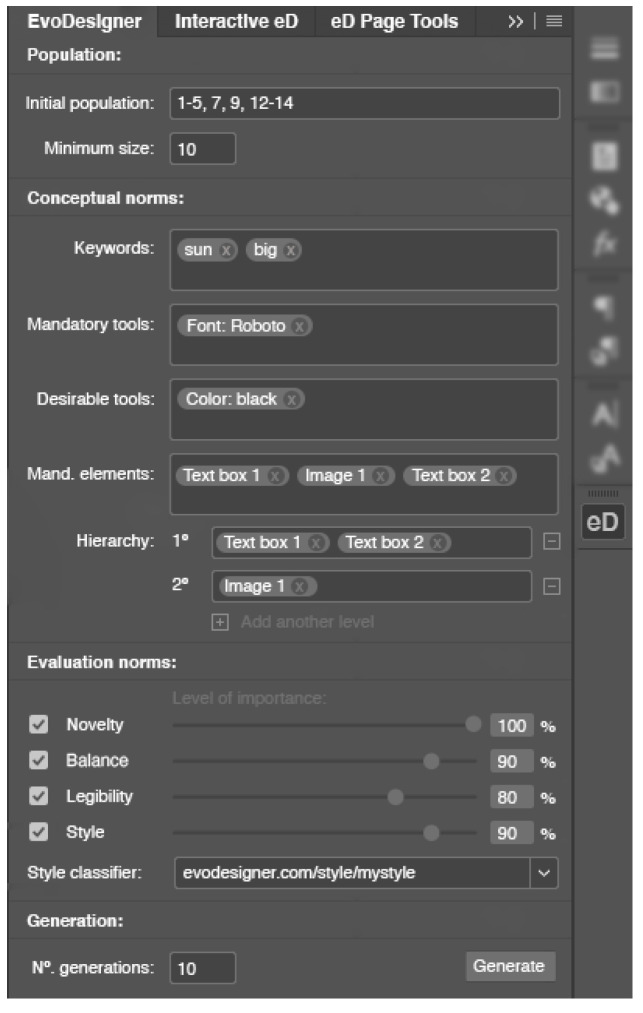
User interface designed to interact with *EvoDesigner* within *Adobe InDesign*. Not all functionalities are in use for the present experiments.

**Figure 3 entropy-24-01751-f003:**
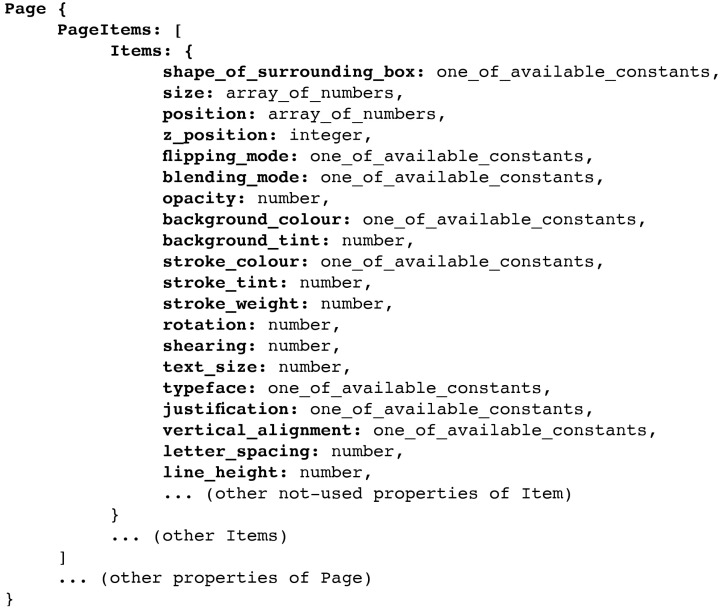
Genotype depiction in schematic form (this scheme serves only for the sake of the example; thus, the property names and value types might not be fully accurate).

**Figure 4 entropy-24-01751-f004:**

Examples of possible page grids the system might create.

**Figure 5 entropy-24-01751-f005:**
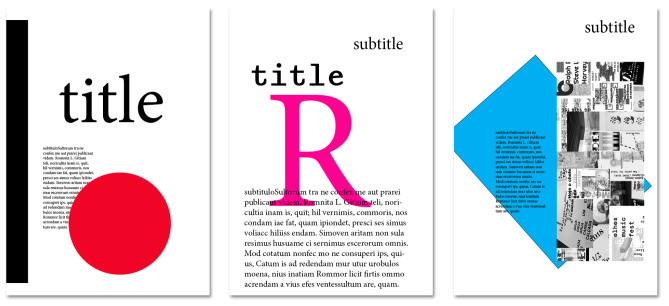
Pages selected to be evolved (manually created from blank pages, lightly stylised).

**Figure 6 entropy-24-01751-f006:**
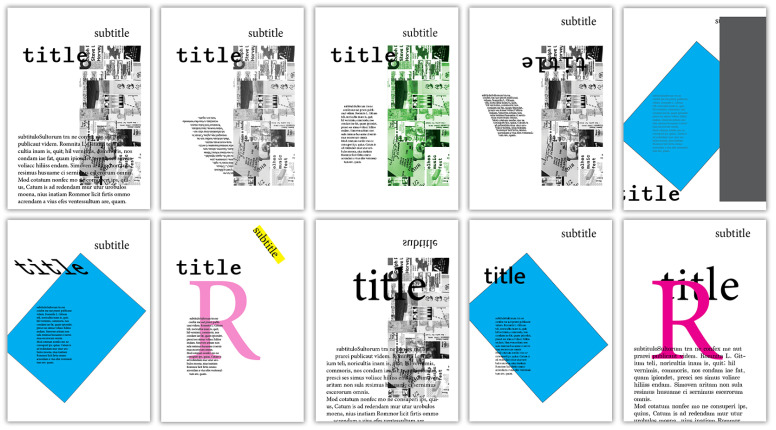
Sample initial population of ten pages (individuals) created from the three selected pages in Figure 5.

**Figure 7 entropy-24-01751-f007:**
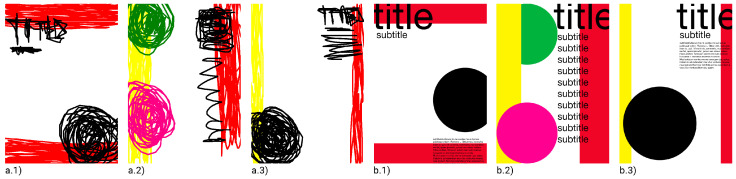
Possible target images: (**a.1**, **a.2**, **a.3**) examples of sketched posters; (**b.1**, **b.2**, **b.3**) examples of camera-ready posters designed in *Adobe InDesign*.

**Figure 8 entropy-24-01751-f008:**
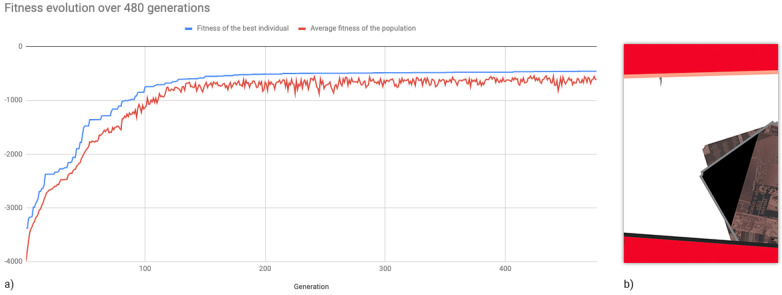
Results from 480 generations using Figure 7b.1 as a target: (**a**) the fitness values of the best individuals of each generation and the average fitness for each population; (**b**) the best phenotype from the 480th generation.

**Figure 9 entropy-24-01751-f009:**
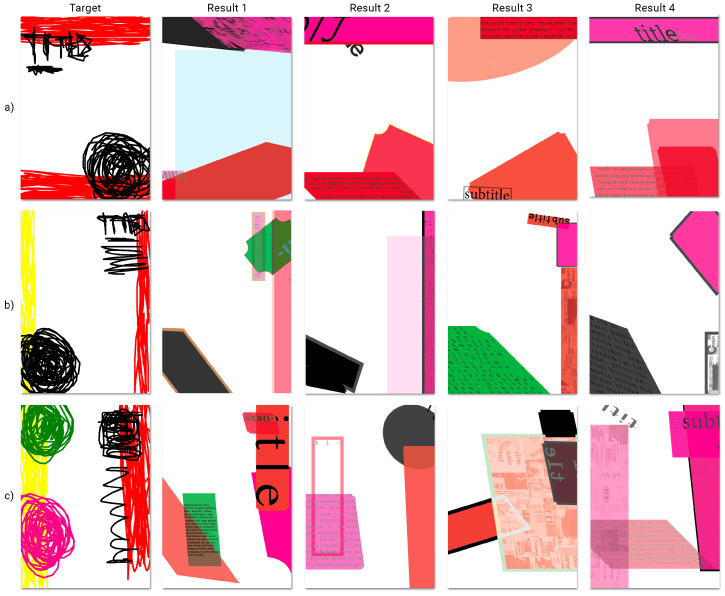
Best individuals from four different runs (100 generations) for three different target images: (**a**) Figure 7a.1; (**b**) Figure 7a.2; (**c**) Figure 7a.3.

**Figure 10 entropy-24-01751-f010:**
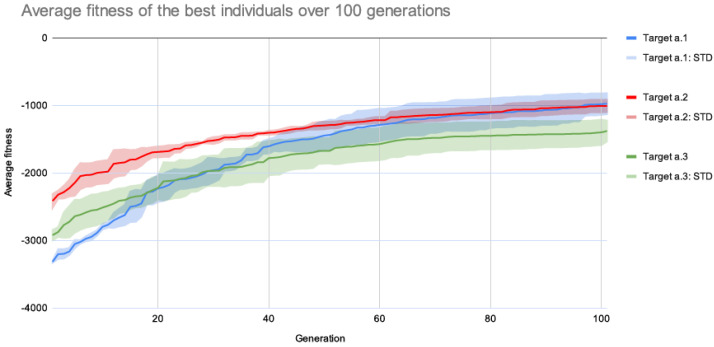
Average fitness (four runs) of the best individuals of each generation for each target image in Figure 7a.

**Figure 11 entropy-24-01751-f011:**
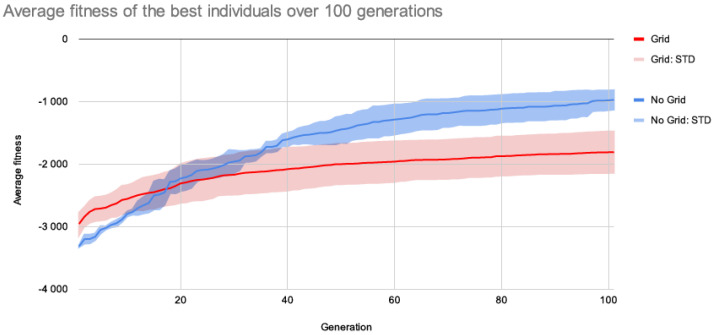
Comparison between the average fitness of the best individuals for the target a.1 using no grids (from previous experiments), and the average fitness of the best individuals using grids for the same target.

**Figure 12 entropy-24-01751-f012:**
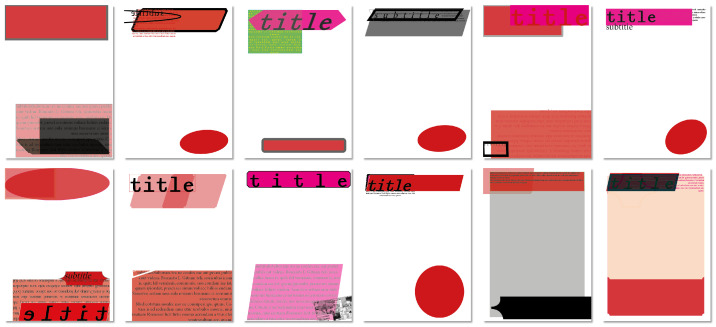
Examples of phenotypes evolved using grids over 100 generations.

**Figure 13 entropy-24-01751-f013:**
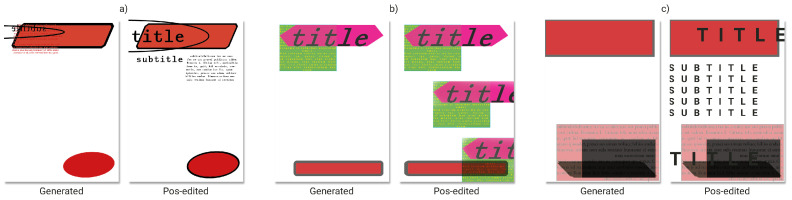
3 examples, (**a**,**b**) and (**c**), comparing automatically generated posters using grids (on the left) and post-edited ones (on the right).

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
