# Peer review of "EvoDesigner: Evolving Poster Layouts"

_entropy, 2022, doi:10.3390/e24121751_

Round 1

Reviewer 1 Report

The paper describes EvoDesigner, an InDesign extension developed by the authors for generating or improving pages and posters layouts. For that purpose, EvoDesigner uses an evolutionary algorithm with automatic fitness assignment.

As stated by the authors, the proposed paper is an extension of a previously published work on the subject. Although the paper is well-written and some interesting results are given, there is not, in my opinion, technical or scientific novelty to justify a new (journal) paper.

I suggest that the authors revise the paper and try to include some of the items addressed in the last paragraph of the last section. Accordingly, they should clearly state the novel contributions of the paper regarding previous publications.

Author Response

As stated by the authors, the proposed paper is an extension of a previously published work on the subject. Although the paper is well-written and some interesting results are given, there is not, in my opinion, technical or scientific novelty to justify a new (journal) paper.

R. In the iteration presented in the present article, we implemented:

  1. New mutation method to create grid systems;
  2. Two other methods so page items can be placed and sized according to the respective grids, promoting more organised layouts;
  3. New experiments on evolving pages using page grids rather than only evolving with no organisation constraints;
  4. We compared the new and old results regarding fitness and phenotypes;
  5. We conducted a user survey to understand whether the newly generated results were indeed more organised than the previous ones, and also to realise whether more organised layouts were a synonym for better designs;
  6. Contrary to the previous paper, we present examples of post-edited posters to demonstrate the possible collaboration of human-machine.

I suggest that the authors revise the paper and try to include some of the items addressed in the last paragraph of the last section. Accordingly, they should clearly state the novel contributions of the paper regarding previous publications.

R. Please, for a description of the novel contributions, refer to the previous reply. Unfortunately, implementing and testing the items addressed in the last paragraph of the last section would take too long to be included in the present publication in time.

Reviewer 2 Report

The article is sound and very well written.
The authors describe in great detail the genotype, phenotype, and evolutionary cycle of the proposed GA algorithm.

My only suggestions are the following:

1) Minor spell check should be done has there are some typos and grammar inconsistencies.

2) Increase the resolution of Figure 3, very low resolution compared to other figures.

3) Explore the diversity of the population. With such a small population and such a large number of generations,  it is possible that some sort of premature convergence and thus lack of diversity is happening. If so, and with such a large search space, it would be an ideal scenario to apply some sort of novelty search to further explore the landscape.

Author Response

Minor spell check should be done has there are some typos and grammar inconsistencies.

R. A further spell check was made using a spell check system.

Increase the resolution of Figure 3, very low resolution compared to other figures.

R. We fixed the resolution issue in Figure 3.

Explore the diversity of the population. With such a small population and such a large number of generations,  it is possible that some sort of premature convergence and thus lack of diversity is happening. If so, and with such a large search space, it would be an ideal scenario to apply some sort of novelty search to further explore the landscape.

R. Thank you very much for the valuable suggestions. Due to present time constraints, we will test larger populations and implement novelty search mechanisms in future work.

Reviewer 3 Report

The paper entitled 'EvoDesigner: Evolving Poster Layouts' is a very interesting manuscript that is an extension of the earlier study. It presents an evolutionary extension for Adobe InDesign. I believe that the structure of the work is correct, the literature review is appropriate and the work adds new value. I only have a few minor comments:

1. I think the reference to literature with '[1]' is unnecessary in the abstract.

2. On page 7, the abbreviation MSE was entered unnecessarily (it was introduced on page 2).

3. The Genetic Algorithm is a nondeterministic method, therefore each run may produce a different result. When testing nondeterministic methods, we should run them at least 10 times and provide basic statistics (at least the average result and standard deviation) to enable the assessment of the impact of nondeterminism on the results obtained. Please add the standard deviation of the results obtained.

4. How were the parameter values in lines 319-324 determined? Have you tested any other parameter values (they are crucial for the effectiveness of metaheuristics)?

Author Response

I think the reference to literature with '[1]' is unnecessary in the abstract.

R. We removed the reference in the abstract.

On page 7, the abbreviation MSE was entered unnecessarily (it was introduced on page 2)

R. We corrected the MSE abbreviation.

The Genetic Algorithm is a nondeterministic method, therefore each run may produce a different result. When testing nondeterministic methods, we should run them at least 10 times and provide basic statistics (at least the average result and standard deviation) to enable the assessment of the impact of nondeterminism on the results obtained. Please add the standard deviation of the results obtained.

R. We added the standard deviation of the results obtained. Thank you very much for the guidance on the statistics issues. We will take them into consideration in further experiments.

How were the parameter values in lines 319-324 determined? Have you tested any other parameter values (they are crucial for the effectiveness of metaheuristics)?

R. The referred parameters were set as in the previous paper so we could properly compare the results. Lines 319-324 describe the setup of the experiments made in the previous paper. In further experiments, we must test different setups. Due to time constraints, it will not be possible to do it in this article.

Round 2

Reviewer 1 Report

After the authors' enumeration of new contributions, I grant that the paper may have sufficient novelty for publication. I suggest that the authors integrate their reply in the last paragraph of section 1, so that the contributions of the paper are clear from the start.